# Unleashing virus structural biology: Probing protein and membrane intermediates in the dynamic process of membrane fusion

Kelly K. Lee 🔵

Department of Medicinal Chemistry, University of Washington, Seattle, WA, USA

**Perspective**

CryoEM structural biology; dynamics and function; integrative structural biology; membranes; virology

**Corresponding author:**
Kelly K. Lee;
Email: kklee@uw.edu

## Abstract

Viruses are highly dynamic macromolecular assemblies. They undergo large-scale changes in structure and organization at nearly every stage of their infectious cycles from virion assembly to maturation, receptor docking, cell entry, uncoating and genome delivery. Understanding structural transformations and dynamics across the virus infectious cycle is an expansive area for research that that can also provide insight into mechanisms for blocking infection, replication, and transmission. Additionally, the processes viruses carry out serve as excellent model systems for analogous cellular processes, but in more accessible form. Capturing and analyzing these dynamic events poses a major challenge for many structural biological approaches due to the size and complexity of the assemblies and the heterogeneity and transience of the functional states that are populated. Here we examine the process of protein-mediated membrane fusion, which is carried out by specialized machinery on enveloped virus surfaces leading to delivery of the viral genome. Application of two complementary methods, cryo-electron tomography and structural mass spectrometry enable dynamic intermediate states in intact fusion systems to be imaged and probed, providing a new understanding of the mechanisms and machinery that drive this fundamental biological process.

## Finding a captivating biological question with room to grow

I was introduced to the concept that viruses can undergo major conformational changes while I was a graduate student in the lab of Bertrand Garcia-Moreno E., where my thesis work centered on precise measurement of electrostatic interactions, dielectric constants, and pH-linked effects in proteins (Lee *et al.*, 2002*a*, 2002*b*). We were focused on small, single-domain proteins such as staphylococcal nuclease, but Bertrand often talked about how changes in charge state and buried ionizable residues with shifted pKa values were found in large, dynamic viral proteins that had evolved to exploit pH-linked effects and environmental signals in order to trigger conformational changes during infection.

Later, during my postdoctoral training with Jack Johnson, I had the opportunity to apply biophysical approaches to study the elaborate, cooperative structural transformations that many bacteriophages undergo during their DNA packaging and maturation events. During this critical stage of assembly, hundreds of protein subunits in the icosahedral capsids reorganize through rotations and translations, expanding like nanoscale Hoberman spheres (Figure 1)(Hoberman, 1988). The increases in volume and stability of the bacteriophage capsids transform the capsid such that the mature assembly is able to encapsulate DNA packaged to pressures rivaling bottled champagne (Roos *et al.*, 2012).

In establishing my independent research program, I searched for a biophysical mechanism and biological system where research at the intersection of structure, dynamics, and infectious disease could address long-standing questions of broad biological significance. I chose to dive into the study of protein-mediated membrane fusion since it is a fundamental biological process that takes place at pivotal stages of enveloped virus infection as well as during cellular vesicle trafficking, neuronal signaling, mitochondrial maintenance, and in cell–cell fusion events that take place during gamete fertilization, placenta morphogenesis, and muscle formation to name just a few examples (Martens and McMahon, 2008; Rey and Lok, 2018). Intriguingly, while the fundamental physical process that must be carried out is the same, namely fusion proteins draw two membranes together, initiate lipid merging, and break down the membrane barrier separating previously membrane-bound compartments, allowing their contents to mix, the machinery to carry out this process has evolved multiple times based upon many distinct protein folds and complexes. In addition, the environmental triggers that activate fusion machinery leading to the conformational changes that drive membrane remodeling are diverse, ranging from pH-induced activation following endocytic uptake into cells, to binding of one or more cellular receptors, to proteolytic cleavage of viral subunits, as well as combinations of these (White *et al.*, 2023). How do diverse machines carry out what appears to be similar physical process of fusing two

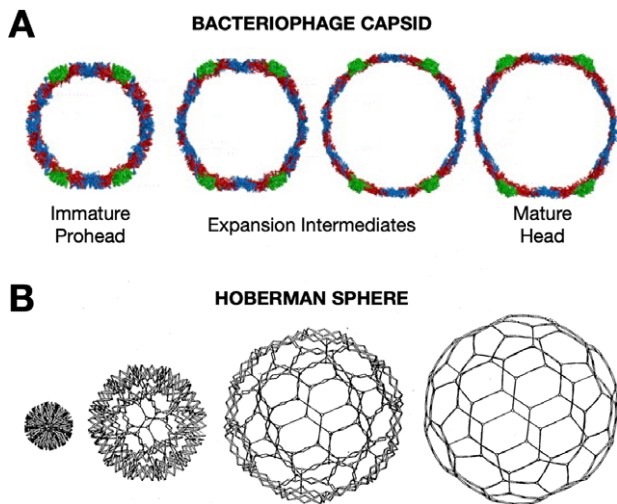

**Figure 1.** Viruses undergo dramatic conformation changes throughout their infectious cycles. **(a)** One example of this is seen in bacteriophage capsid maturation, where hundreds of protein subunits organized in an icosahedral shell reorganize while maintaining capsid integrity to produce an expanded mature capsid with thinner but more stable walls—due to greater inter-subunit contacts and fewer holes in the lattice—and greatly increased internal volume. A cross-section of HK97 bacteriophage is shown. Figure adapted from Wikoff *et al.* (2006)). This structural transformation is highly cooperative and is highly reminiscent of the expansion of **(b)** a mechanical Hoberman sphere, a "reversibly expandable, double-curved truss structure" Figure adapted from USA Patent US4942700A (Hoberman, 1988).

membranes into one, and do they traverse similar intermediate states and follow similar reaction pathways?

To address these questions, one would like to image and understand how fusion protein conformational change is coupled to membrane reorganization and to identify what stages of membrane bilayer deformation are populated leading to lipid mixing and fusion. While it had long been speculated that an aqueous fusion pore is ultimately formed following fusion, it had not been clear what its architecture might be and what steps of membrane remodeling lead up to its formation. The prevailing hypothesis that emerged from a rich body of theoretical and modeling studies suggested that fusion initiates when fusion proteins induce localized dimple-like contacts, leading to point-like interfaces that result in merging of the proximal outer leaflets forming a "hemifused stalk" (Figure 2) (Kozlov and Markin, 1983; Kozlovsky *et al.*, 2002; Chernomordik and Kozlov, 2005). It had been proposed that the stalk then converts into a widened hemifused bilayer, or diaphragm, composed of the two distal or inner leaflets, which subsequently disintegrates to open a fusion pore allowing content transfer (Chernomordik *et al.*, 1995; Kozlovsky *et al.*, 2002; Nikolaus *et al.*, 2011).

At the time it was not possible to directly test these hypothetical models, though extensive experimentation using fluorescence spectroscopy and microscopy could track the kinetics of lipid and content mixing. These early findings provided indirect support for the proposed stages of membrane reorganization but lacked the sufficient resolution needed to visualize detailed membrane structure and individual fusion proteins. Indeed, even superresolution light microscopy can only resolve fluorescent probes at ~20 nanometer separation (Bond *et al.*, 2022), whereas the phosphate headgroup layers for the leaflets of a membrane bilayer are typically spaced ~3–4 nm apart. Moreover, at the time, mechanisms of protein conformational change and their action on membranes were largely inferred from stable, inert pre- and post-fusion structures of ectodomain fragments that lacked key structural elements such as transmembrane anchors, membrane, and cytoplasmic domains that modulate fusion protein activity. The protein states that drive fusion reactions, however, are generally refractory to classical structure determination due to their transience and flexible, dynamic nature. Methods capable of capturing and imaging protein-mediated fusion reactions for intact fusion systems at molecular resolution were needed in order to reveal fusion mechanisms and to characterize critical intermediate states.

## Choosing effective techniques to address the question

As it turns out, around the time I was learning about conformational changes in viruses, imaging techniques that were beautifully suited to address the questions of how viral machinery mediates membrane fusion were coming online and ready for application. Cryo-electron microscopy (cryo-EM) has revolutionized structural biology by enabling all manner of biological systems to be imaged in their native hydration states without requiring crystallization and fixation. Structure determination of complex macromolecular assemblies was unleashed from the constraints of crystallization. While structures of ribosomes, icosahedral viruses, and large macromolecular assemblies demonstrated the power of single particle analysis approaches, this approach generally involves stringent classification of individual images of each macromolecule or complex, sorting out conformational heterogeneity and variation, followed by averaging over tens and hundreds of thousand images of the most uniform computationally purified samples (Cheng, 2015; Cheng *et al.*, 2015; Chari and Stark, 2023). The resulting high-resolution structures in some cases, can be based upon a small percentage of the total input dataset. Tomographic approaches by comparison offer a means to obtain 3D structures of complex biological objects that may not be amenable to global averaging. With cryo-electron tomography (cryo-ET), one images the sample over a range of axial angular tilts, then reconstructs the individual

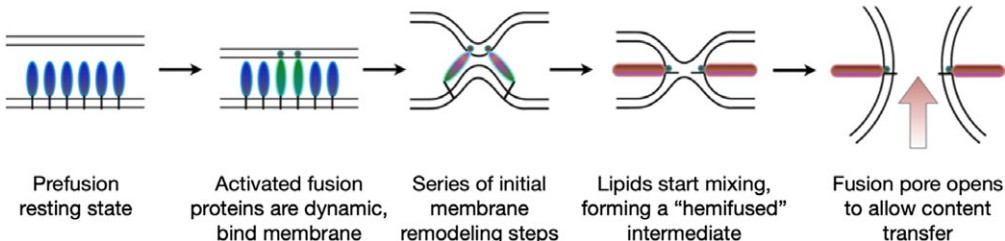

**Figure 2.** A common pathway has been proposed for protein-mediated membrane fusion leading to fusion pore formation via a "hemifused" intermediate.

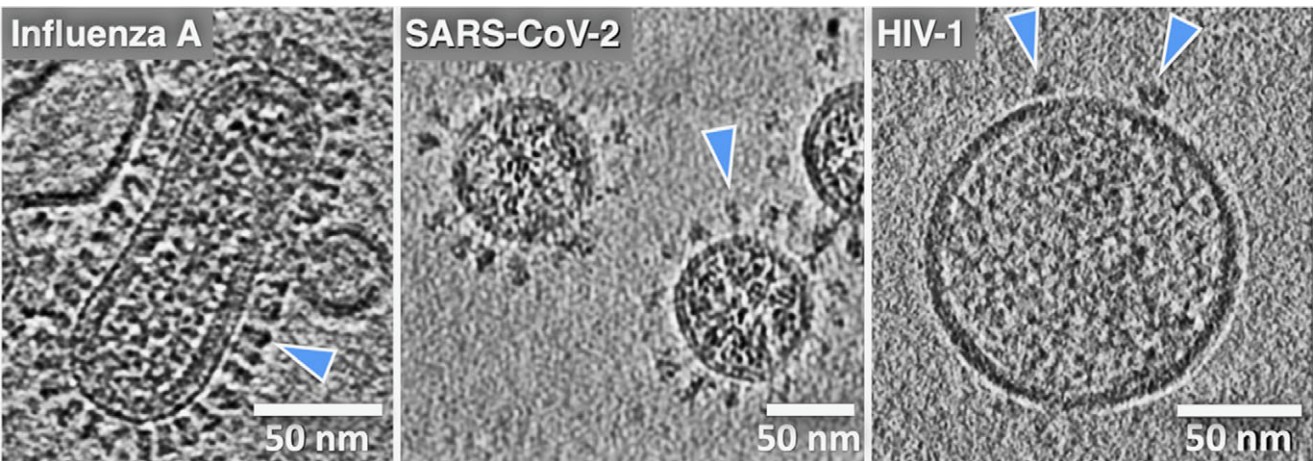

**Figure 3.** Cryo-electron tomography reveals virus ultrastructure with resolution of individual fusion glycoproteins (blue arrows), membrane organization, and internal viral contents. **(a)** Central section for influenza A virus, **(b)** paraformaldehyde fixed SARS-CoV-2, and **(c)** HIV-1 virus particles. The differences in fusion protein abundance and distribution are clearly evident. (cryo-ET by Drs. Long Gui, Nancy Hom, and Vidya Mangala Prasad).

images into a 3D tomogram without relying on averaging over populations of conformationally homogeneous targets (Ognjenovic *et al.*, 2019).

One common theme that has motivated our research and that seems to have been borne out by experiments is that it is difficult to infer how viral machinery or any biological machinery functions when examining just fragments or isolated components. When excised from their native membrane-bound contexts, the fusion proteins can lose stability, alter their conformation, and shift their dynamic behavior and activity. The ability to analyze whole viruses and intact, functional fusion systems using tomography has been critical to the effort to apply cryo-EM to analyze membrane fusion reactions. Indeed, viruses were some of the first complex biological objects visualized using cryo-ET, starting with herpes simplex, vaccinia, and later influenza A virus, HIV-1, SIV, and others (Figure 3) (Li, 2022). For the first time, using cryo-ET, the native architecture of whole intact viral particles could be resolved with molecular detail. Cryo-ET is the only structural technique available that can be used to image the 3D organization, abundance, and disposition of viral surface proteins, as well as to visualize internal structural assemblies such as matrix and capsid layers, nucleoprotein complexes, and membrane fine structure, including local curvature and leaflet integrity. Cryo-EM/ET can even be used to directly image lipid nanodomains (Cornell *et al.*, 2020; Heberle *et al.*, 2020). Of note, cryo-ET is also an especially powerful technique for studying biological processes such as membrane fusion because one can trigger a reaction and then flash-freeze the specimen, trapping intermediates and allowing the 3D organization of the fusion complexes to be visualized. It is thus perfectly suited to reveal how viral machinery drives membrane fusion.

## Selecting a biological system to focus on

With an exciting biological question and a suitable experimental technique in hand, next it was necessary to choose a tractable system to study. As a virus and as a model system for membrane fusion, the influenza virus has been studied intensively for decades. Influenza hemagglutinin (HA), the virus' receptor binding and membrane fusion protein, an archetypal largely helical, trimeric class-I fusion protein, was the best characterized system where crystal structures were available for pre- and post-fusion conformations of the isolated

ectodomain, and considerable biophysical information was available from numerous techniques such as fluorescence spectroscopy and microscopy, NMR, negative stain EM, and even thin-section TEM of virus entering into cells (Figure 4a) (Matlin *et al.*, 1981). Cryo-ET also had been shown to be effective for visualizing the architecture of whole influenza virions by Alasdair Steven's and later Peter Rosenthal's labs (Harris *et al.*, 2006; Calder *et al.*, 2010).

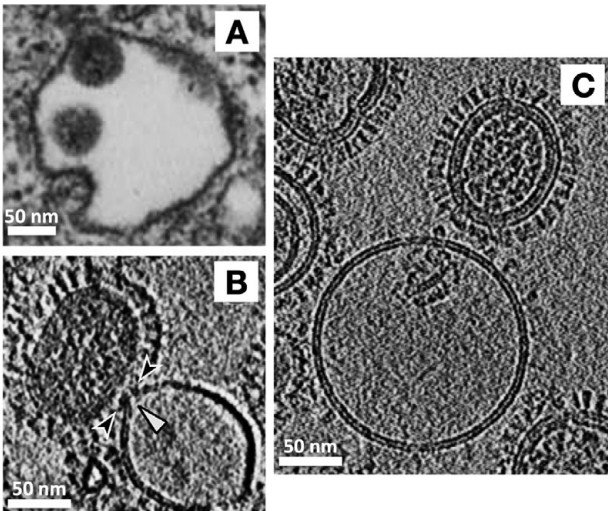

**Figure 4.** Advances in resolution of influenza virus-membrane interactions. **(a)** Some of the earliest glimpses of influenza's membrane fusion event were captured for the virus inside of endosomes using heavy metal-stained, thin-section TEM of fixed cells (Matlin *et al.*, 1981). The pathway of virus entry could be tracked, but direct resolution of protein and membranes and their structures was not possible with this approach. With the application of cryo-ET, a virus interacting with liposomes *in vitro* could be visualized under native buffer conditions in vitrified ice. Over the past several years, technical advances have increased resolution and data quality as is evident from the sharper images of proteins as well as membrane fine structure including resolution of leaflets in cryo-electron tomograms. For example **(b)** shows a central section through a tomogram of influenza virus interacting with a liposome that was collected using a cryo-EM configuration with a side-entry sample holder and charge-coupled device (CCD) (Lee, 2010), while **(c)** shows a tomogram collected on a contemporary microscope with a more sensitive, fast frame readout direct electron detector, an energy filter to filter out inelastically scattered electrons, and a more stable specimen stage (cryo-ET image provided by Dr. Vidya Mangala Prasad).

The influenza virus was also an advantageous system for *in vitro* studies because the trigger that activates its HA fusion machinery is low pH, rather than receptor binding as is often employed by other viruses. Influenza encounters this activation signal following endocytic uptake into cells when endosomal maturation leads to staged drops in pH to ~5, which triggers HA's membrane fusion activity, leading to genome delivery into the cytosol. *In vitro*, it was relatively straightforward to use liposomes and vesicles as a stand-in for the endosomal membrane and to acidify the buffer to mimic endosomal pH conditions (Figure 4b,c). Such an experimental setup also offered considerable control over target membrane lipid composition, pH, temperature, and timing, enabling us to test the influence of those variables on the fusion mechanism.

With this approach, we and others were able to capture snapshots of the intermediate states that the influenza virus traverses across the fusion pathway (Lee, 2010; Calder and Rosenthal, 2016; Chlanda *et al.*, 2016; Gui *et al.*, 2016; Benhaim *et al.*, 2020). Moreover, by carrying out a time course and tracking the ebb and flow of intermediate state populations as the reaction proceeds, it was possible to infer the sequence that the reaction intermediates followed and reconstruct the pathway leading from fusion protein activation to stages of membrane remodeling all the way to fusion pore opening (Gui *et al.*, 2016). These studies demonstrated that the initial stages of membrane remodeling, driven by hairpin refolding of a localized set of HA trimers, were almost entirely focused on the target membrane, resulting in the formation of tightly curved dimples that are drawn down to the relatively unperturbed virus surface. These studies demonstrated the role of the internal, membrane-associated viral matrix layer in butressing the viral envelope during these early stages of membrane reorganization. The matrix layer also exhibits pH-dependent activity, dissociating from the viral membrane at a pH below HA's activation pH (Fontana and Steven, 2013; Gui *et al.*, 2016). Thus, the internal superstructure helps to regulate viral membrane plasticity and progression along the fusion pathway. Indeed, the fusion machinery for influenza virus, and it is likely for many enveloped viruses that bear structural membrane-associated matrix layers, is embodied by not just the fusion protein itself, but the fusion protein, the viral membrane, and matrix proteins working in concert.

Strikingly, we observed that following dimple formation, rather than observing transitions to the previously proposed hemifusion stalk organization (Figure 2), the target and virus membranes became tightly docked over extended areas in which the proximal leaflets were apparently so closely apposed that they often appeared superimposed (Figure 5). This lipid configuration has also been observed in SNARE-mediated and mitofusin-mediated fusion

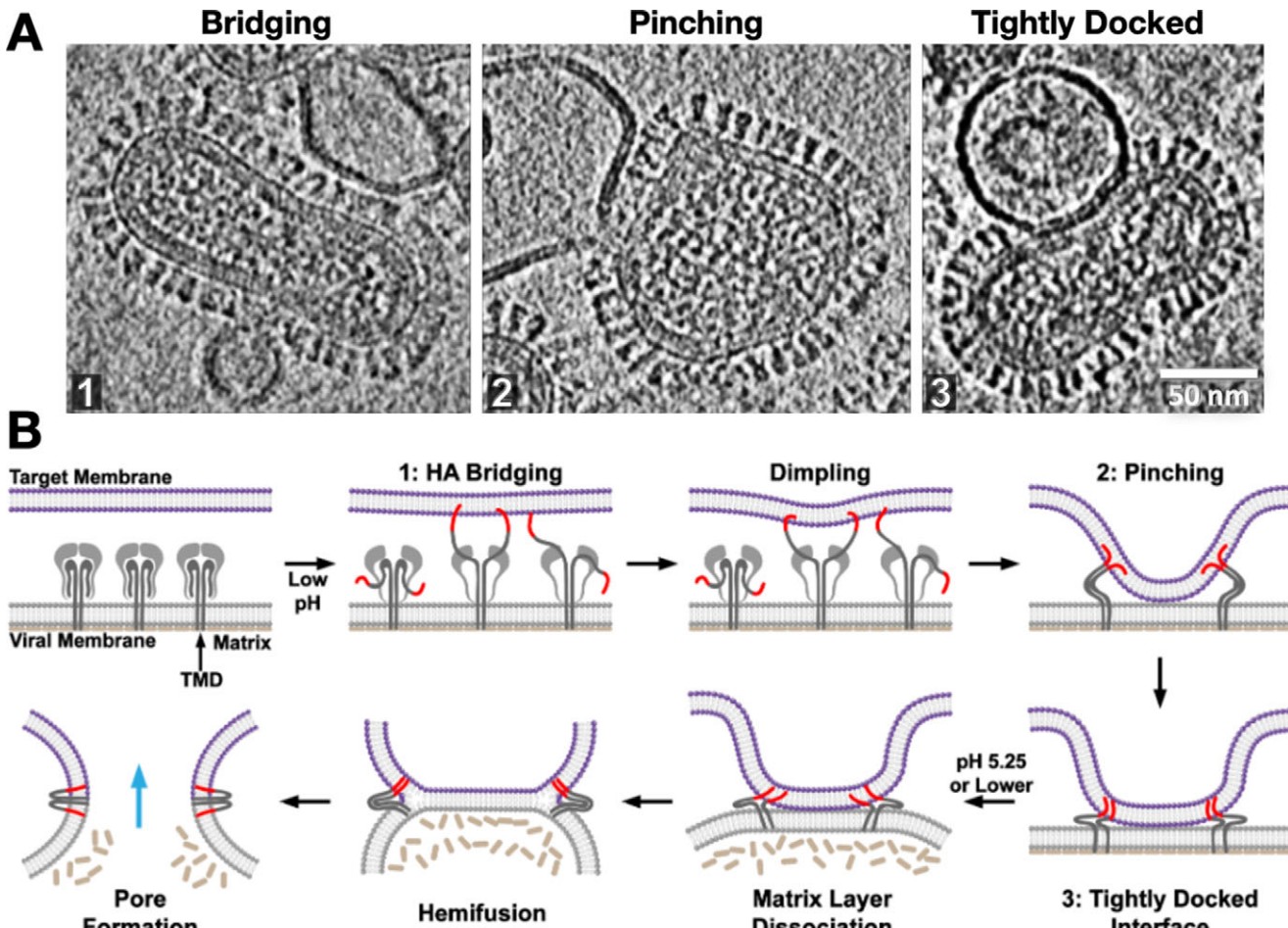

**Figure 5.** Snapshots and inferred pathway of influenza virus HA-mediated membrane fusion with liposomes visualized with cryo-ET. **(a)** Several intermediate membrane remodeling intermediates can be captured by initiating fusion reactions in vitro and flash-freezing the reaction (Figure adapted from (Benhaim and Lee, 2020), original data in (Gui *et al.*, 2016)). **(b)** By carrying out a time course and monitoring the population kinetics of these intermediates, the sequence and pathway for HA-mediated fusion were inferred. These experiments highlighted the centrality of the tightly docked membranes following target membrane apposition to the virus surface through localized dimples but did not capture classical hemifused "stalks." Cryo-ET also demonstrated that the internal matrix structural layer works in concert with HA to mediate fusion.

complexes (Mirjanian *et al.*, 2010; Hernandez *et al.*, 2012). Interestingly simulations suggest that headgroups can line up along a single layer, appearing as a single leaflet, while acyl chains are splayed across the two membranes (Stevens *et al.*, 2003; Kasson *et al.*, 2010) or alternatively, the largely dehydrated headgroups from proximal leaflets may interact with each other to in essence "solvate" each other through dipolar coupling, reducing the dehydration penalty that is typically associated with bringing two phospholipid layers into close proximity (Witkowska *et al.*, 2021). Tightly docked membrane contacts increased in abundance with the addition of fusion-promoting lipids and rose to be the dominant intermediate after dimple formation during influenza virus fusion (Gui *et al.*, 2016). These intermediates then dissipated as post-fusion complexes rose, suggesting they lie on the fusion pathway. Surprisingly, in our hands, one of the main membrane fusion intermediates that had been previously investigated, the hemifusion diaphragm, was observed to be exceedingly rare, perhaps indicative of its transience and instability in the influenza virus system (Gui *et al.*, 2016), while other groups were able to arrest fusion in hemifused states using HA fusion peptide mutants (Chlanda *et al.*, 2016).

### Generalizing studies to identify conserved properties and fundamental physical aspects of the biological process

Visualization of membrane fusion intermediates at the level of detail described above would not have been achievable using any other biophysical or imaging approach. Cryo-ET analysis of numerous protein-mediated membrane fusion reactions with different fusion protein systems have now revealed striking commonalities, even for viruses from different families that employ protein fusogens from distinct structural folds organized on the virus surface in drastically different ways (White *et al.*, 2023; Kephart *et al.*, 2024). For example, class-II fusion proteins, such as those in flavi- and alphaviruses as well as in some eukaryotic gamete and cell fusion systems, are based upon an entirely different fold from the class-I fusogens (Rey and Lok, 2018). Despite this basic difference, these fusogens also have a membrane anchor and a fusion loop that is initially sequestered, only to become unmasked in response to a trigger. Once unmasked, the fusion loop inserts into a target membrane. Like in class-I fusion proteins, a switch to a folded-back hairpin is believed to colocalize the membrane-active elements, drawing the membranes together.

Perhaps not surprisingly, initial stages of the membrane fusion reaction in which the target membranes are engaged differ rather significantly between systems such as influenza HA and alphavirus E1-mediated reactions that have been described to date (Cao and Zhang, 2013; Calder and Rosenthal, 2016; Chlanda *et al.*, 2016; Gui *et al.*, 2016; Mangala Prasad *et al.*, 2022). However, once the fusion proteins fold back to draw the membranes into apposition, common membrane and fusion protein configurations are observed. Namely, an increasing abundance of experimental observation seems to indicate that tightly docked membranes with juxtaposed or even comingling proximal leaflets are populated by nearly all fusion systems observed to date.

Later stages of fusion reveal some differences, particularly in the propensity for the fusion complexes to transition between the states we have observed, such as between tightly docked membranes, hemifusion diaphragms, and open fusion pores. For example, while hemifusion was a rare, transiently populated intermediate with influenza virus, for alphavirus fusion, large hemifused diaphragms that spanned nearly the viral particle diameter were relatively abundant at intermediate time points, and then dissipated as the reaction proceeded (Mangala Prasad *et al.*, 2022). The diaphragms were seen to disintegrate, opening fusion pores that enabled nucleocapsid delivery. The greater instability of hemifusion diaphragms among class-I viral fusion systems may, in part, be due to the ability of transmembrane and fusion peptide domains to form tight complexes in the folded-back hairpin state, which likely applies force to the membranes and destabilizes local lipid packing. By contrast, such a coupling of the two membrane active motifs, fusion loop and transmembrane domain, in class-II fusion proteins does not seem to be generally observed.

### What do the cryo-ET snapshots miss?

One notable limitation of the "time-resolved" cryo-ET approach is it mainly reveals the metastable states that accumulate in population over the course of a fusion reaction, but the transient transition states are sparingly and fleetingly populated at any given time and thus difficult to capture and characterize in detail. To have a hope of analyzing these transient intermediates will require more extensive sampling of larger populations. Computational modeling, molecular dynamics and simulation coupled to quantitative analysis of cryo-ET data can also provide highly detailed insights into the transitions between membrane configurations and the forces and energetics that shape membrane remodeling during fusion.

On a related note, the essential dimension of time is missing from the cryo-ET snapshots. Population kinetics can show how populations of viruses progress along a reaction pathway, taking seconds to minutes to reach completion, but each individual reaction thus far can only be monitored using fluorescence light microscopy and tracking the exchange of lipid probes and fluorescent contents, and this appears to occur on the millisecond-second time scales (Floyd *et al.*, 2008; Blijleven *et al.*, 2016).

### Turning to complementary methods to probe protein dynamics and conformational change

As informative as the cryo-ET analysis of protein-mediated membrane fusion has been, particularly for revealing membrane remodeling states and how fusion proteins engage with the membranes, complementary methods are needed to probe how the fusion protein machinery itself is acting during the fusion process. Cryo-ET at present does not provide sufficient resolution to reveal how proteins are refolding beyond highlighting large-scale changes in structure as the fusogens drive membrane remodeling, though coupled with modeling and simulations, quasiatomic models can be compared with and docked into the tomograms.

Given the dynamic nature of the fusion machinery, our lab sought to develop experimental methods that would enable conformational changes to be monitored in the protein machinery itself. Various approaches to investigate viral protein dynamics have been described in recent years including, for example, single-molecule FRET and molecular dynamics simulations (Risselada and Grubmuller, 2021; Groves *et al.*, 2023). We pursued a structural mass spectrometry approach called hydrogen/deuterium-exchange mass spectrometry (HDX-MS) (Hodge *et al.*, 2020). HDX-MS probes the degree to which backbone amides are involved in secondary structure and indirectly in deeply buried regions of protein (Venkatakrishnan *et al.*, 2024). HDX-MS does not provide 3D structural information but rather enables us to track local protein dynamics along segments of polypeptide sequence. It is particularly effective for probing conformational changes, effects of

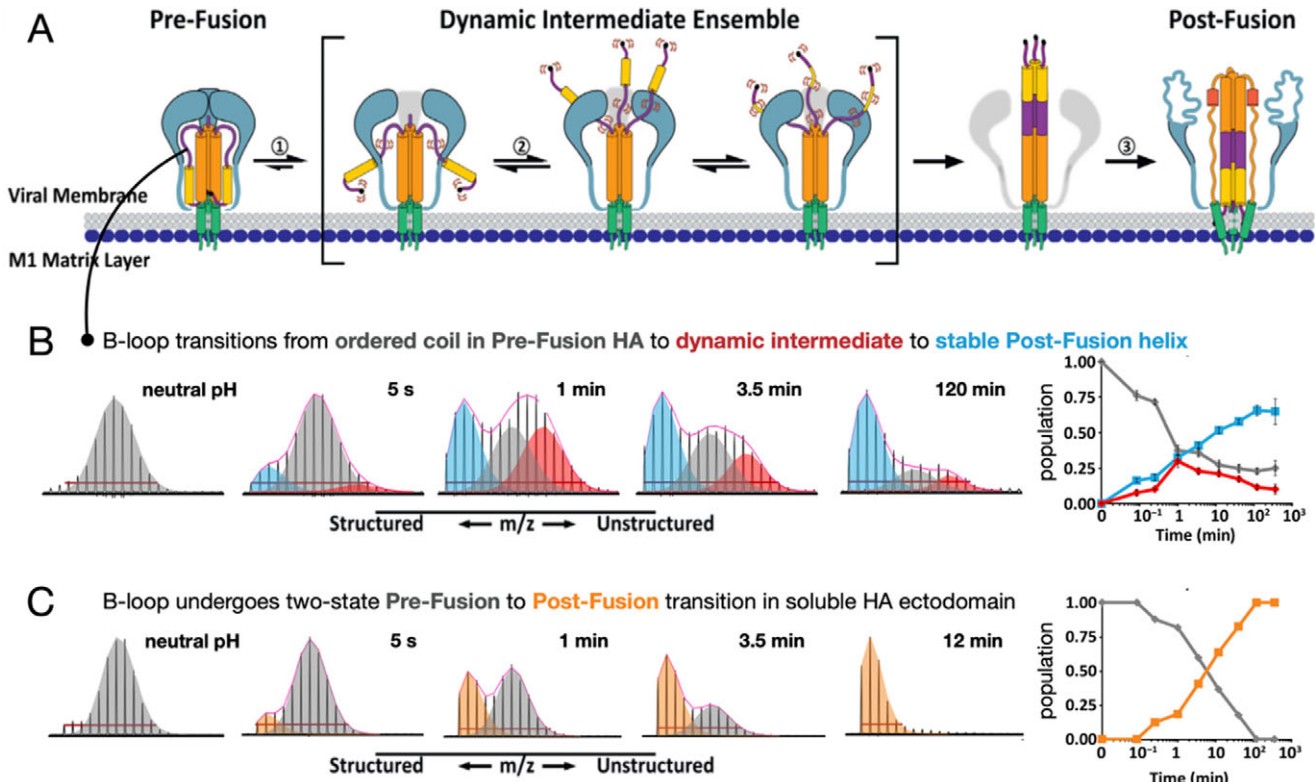

**Figure 6.** Fusion proteins in native contexts, such as on whole virions, function differently than isolated, soluble ectodomains that are typically the subject of structural studies. We used pulse deuteration HDX-MS to monitor activation and conformational change of influenza HA refolding pathway **(a)** on intact influenza virus **(b)** revealing the existence of a transient but highly populated dynamic intermediate state with exposed fusion peptides and dynamic B-loop motifs, in contrast to soluble HA ectodomain **(c)** that converted directly from pre- to post-fusion states (Benham *et al.*, 2020). Pulse deuteration HDX-MS is thus a powerful means to take snapshots as the proteins traverse their pathways of conformational change and track the states of each peptide segment throughout the protein populates over time (insets on the right).

ligand binding, and for examining how proteins respond to changes in the environment.

We and others have used HDX-MS to monitor receptor and environment-triggered conformational changes in viral fusion proteins such as HIV-1 Env, SARS-CoV-2 S, and influenza HA (Figure 6). This method can be applied to both soluble, recombinant proteins, as well as viral fusion proteins in their native membrane-displayed contexts (Davenport *et al.*, 2013; Guttman *et al.*, 2014, 2015; Garcia *et al.*, 2015; Lim *et al.*, 2017; Benhaim *et al.*, 2020; Hodge *et al.*, 2022, 2023; Mangala Prasad *et al.*, 2022b; Braet *et al.*, 2023; Calvaresi *et al.*, 2023). In one study, we applied a pulse deuteration approach to compare pH-triggered refolding of influenza HA in soluble ectodomain form and on intact virions (Benhaim *et al.*, 2020). By labeling the samples with pulses of buffer made with $D_2O$ at specific time points following pH-triggered HA fusion activation, it was possible to track the regions of HA that changed conformation and to monitor how their local structural dynamic behavior changed as they transitioned from pre- to post-fusion conformations.

Early in their study, HA and other viral fusion proteins were described as undergoing a spring-loaded transition from pre- to post-fusion states (Carr and Kim, 1993). Indeed, when one examines isolated fusion protein ectodomains, they often do behave in a two-state fashion, transitioning from a pre- to post-fusion conformation as our pulse HDX-MS analysis of HA demonstrated (Figure 6) (Benhaim *et al.*, 2020). However, when HA is activated in its native context on viral particles, with transmembrane anchor embedded in the viral membrane and cytoplasmic tail present where it can interact with the internal matrix layer, the fusion assembly populates highly dynamic intermediate conformational ensembles with unrestrained fusion peptide domains that can persist for extended periods of time before the assembly ultimately converts to the post-fusion conformation. This is in striking contrast to what was observed for soluble HA ectodomains. The stabilization of dynamic intermediate conformations with fusion peptides freely deployed likely facilitates virus engagement with target membranes.

HDX-MS has also proven uniquely capable of providing structural and dynamic insights into how a given fusion protein assembly differs among strains of a given virus (Davenport *et al.*, 2013; Garcia *et al.*, 2015; Guttman *et al.*, 2015; Lim *et al.*, 2017, 2021; Sharma *et al.*, 2019; Hodge *et al.*, 2022, 2023 ; Braet *et al.*, 2023; Calvaresi *et al.*, 2023). A motivating hypothesis for us was that much of the structural basis for antigenic, phenotypic and functional variation between viral strains would be most evident at the level of structural dynamics, when the virus entry and fusion machines are fluctuating with thermal energy under native conditions. Indeed, the same machines that mediate the pivotal first steps of virus infection of cells, receptor binding, and membrane fusion, are also the primary target for neutralizing antibodies and thus subject to intense immune selective pressures. They undergo rapid diversification and immune escape while maintaining conserved receptor binding and fusion activities. HIV-1 offers an extreme example of such diversification, yet to date, structural analysis has been largely focused on a few well-behaved, stable HIV-1 Env glycoprotein variants. Combined HDX-MS and antibody binding

studies of HIV-1 Env trimers revealed that differences in antigenic phenotype can be traced back to differences in local epitope dynamic fluctuations in the fusion protein assemblies as well as to differences in large-scale conformational dynamic switching where a polypeptide segment and associated structural domain may sample conformational states with different backbone amide protection and deuteration propensities on a relatively slow time scale (Weis *et al.*, 2006; Hodge *et al.*, 2022, 2023; Venkatakrishnan *et al.*, 2024). Likewise, HDX-MS has revealed differences in fusion protein activation dynamics between strains in influenza HA, coronavirus spike proteins, and among dengue virus serotypes (Lim *et al.*, 2017; Sharma *et al.*, 2019; Braet *et al.*, 2023; Calvaresi *et al.*, 2023; Chen *et al.*, 2023; Garcia *et al.*, 2023). Structural dynamics thus embody cryptic determinants of virus phenotype and provide a unique window into strain-specific variation that is difficult to access through most structural approaches.

## Outlook

One important question that looms over the studies described above is: do these processes operate in cells and *in vivo* in ways that track with the *in vitro* studies? Addressing this is becoming more feasible with the availability of cellular tomography methods that utilize focused ion beam milling to produce thin "lamella" suitable for cryo-ET, and recent studies have begun to catch glimpses of virus processes, including membrane fusion deep inside of cells (Graham and Zhang, 2023; Hong *et al.*, 2023; Zimmermann and Chlanda, 2023; Zhou and Lok, 2024). While methods are being developed for in-cell HDX-MS, the method is inherently a bulk population measurement, and it is not feasible to extract information for the subset of fusion proteins that may be mediating fusion at a given time against the overwhelming background of the fusion proteins present in total. Nevertheless, looking ahead, it seems clear that through a combination of *in vitro* and in cell methods, coupled with computational modeling to connect experimentally observed snapshots, the mechanisms that drive the essential and ubiquitous biophysical and biological process of protein-mediated membrane fusion will come into ever sharper focus.

**Open peer review.** To view the open peer review materials for this article, please visit http://doi.org/10.1017/qrd.2025.3.

**Acknowledgments.** The aim of this Perspectives review is to describe one researcher's scientific trajectory rather than to provide a comprehensive overview of protein-mediated membrane fusion or enveloped virus structural biology, which have been reviewed in a number of excellent recent articles (Li, 2022; White *et al.*, 2023; Winter and Chlanda, 2023; Jahn *et al.*, 2024; Kephart *et al.*, 2024; Zhou and Lok, 2024). Studies of viral glycoprotein structure, dynamics, and membrane fusion in the Lee lab are supported by R01-AI165808, R01-AI140868, and R01-AI179697.

**Author contribution.** The author conceived of and wrote this Perspectives review.

**Competing interests.** The author declares that he has no competing financial interests.

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
