## [Reviewer Report]

Kelly Lee has provided us with a survey of the broad history and current state of experimental affairs (cryoEM tomography and hydrogen deuterium exchange (HDX)) of events associated with membrane fusion. He opens with a broad description of the electrostatic effects on virus particle maturation and follows with its impact on membrane fusion associated with enveloped viruses as well as cellular events that require exchange of contents in different membrane compartments. The main focus is on influenza virus hemagglutinin (HA) mediated membrane fusion employing model systems amenable to triggering at a particular moment through lowering the pH of the system. The most recent cryoEM tomograms from the author’s lab provide extraordinary detail of this process and change the view of the frequency of the so-called hemi fusion intermediate, suggesting that it is transient on time scales that make it unlikely to be observed with this method. CryoEM tomographs comparing mechanisms of type I (HA) and type II (e.g. alphaviruses) fusion proteins demonstrate remarkably similar intermediates in spite of the proteins having totally different structures supporting the independent recurrence of the fusion process in evolution. While the overall processes of the type I and type II fusion events are closely similar, the hemi fusion state is significantly more populated in the type II cryoEM tomograms observed to date than type I.

The second method employed by Lee and colleagues and briefly reviewed here is HDX. The example described in some detail is the time resolved HA transition that initiates membrane fusion. Based on ecto-domain HA structures of pre and post fusion states the working hypothesis was that the pre fusion state is “spring loaded” to transition rapidly to the post fusion state. When the soluble ecto-domains (in the absence of membrane interacting regions and membranes) is analyzed by time resolved HDX, the HA transition moves rapidly from one conformation to the other as suggested by the structures. However, time resolved HDX applied to the whole protein in membranes, where fusion is actually occurring, argues against this, displaying a smooth transition of HA between the two states in the time frame of minutes. Lee argues convincingly that it is essential to have all components of the system in place to get a realistic mechanistic picture of these events. Certainly, the model system developed for HA by Lee and colleagues has gone a long way to accomplish this.

The future of this work is to observe these processes in cells employing focused ion beam milling and cryoEM tomography. Lee and colleagues appear well positioned to bring their methods into cells.

This is a very well written and highly informative review and is an excellent contribution to QRB. It should be published as it.

---

## [Reviewer Report]

This is an excellent review that focuses on describing the challenges and advances in membrane fusion of enveloped viruses with host cell membranes. This is an important frontier area and the author has detailed the challenges and advances in Cryo-Electron Tomography (Cryo-ET) and structural mass spectrometry. The author also details their career trajectory that led them to this important research question and model system.

I do have suggestions for enhancing accessibility of the content to readers. These are:

1) It would be suitable for the author to reduce some of the unnecessary hyperbole. In the first sentence, the author describes viruses as ‘exceptionally’ dynamic macromolecular assemblies. I think dynamic (without exceptional)would suffice. Viruses can undergo ‘majpr conformational changes but to describe the changes as ‘gloriously elaborate’ seems quite unnecessary.

2) The title should be amended from “Unleashing virus structural biology”- not clear what unleashing meant. This is more a highly dynamic process.

3) In the legend for figure 1, it is not clear how thinner walls of an expanded mature capsid are more stable. This seems to be subtly contradictory, and so it would be good to qualify better.

4) The review would benefit from additional description on why cryo-EM offered better insights than fluorescence microscopy.

5) The author should describe the limitations of particle picking in single molecule cryo-EM and the limitations of symmetric averaging in inaccurately capturing the essential heterogeneities of virus particles in solution. The author has noted that the notion of time is missing in “time-resolved cryo-ET approach”.

6) Figure 4 excellently captures the power of cryo-ET in advancing description of virus membrane interactions.

7) The author provides a good segue for application of structural mass spectrometry, most notably HDX-MS for studying virus membrane fusion. It is incorrect to assume that HDX-MS only measures solvent accessibility. When it primarily measures H-bond propensities (Englander and Kallenbach, QRB 1983). In this regard, authors should include a citation Venkatakrishnan V, Braet SM, Anand GS. Dynamics, allostery, and stabilities of whole virus particles by amide hydrogen/deuterium exchange mass spectrometry (HDXMS). Curr Opin Struct Biol. 2024 Jun;86:102787. doi: 10.1016/j.sbi.2024.102787. Epub 2024 Mar 7. PMID: 38458088.

8) The authors should describe the advantages of EX1 kinetics of deuterium exchange in uncovering dynamics of virus particles, as described in Lim XX, Shu B, Zhang S, Tan AWK, Ng TS, Lim XN, Chew VS, Shi J, Screaton GR, Lok SM, Anand GS. Human antibody C10 neutralizes by diminishing Zika but enhancing dengue virus dynamics. Cell. 2021 Dec 9;184(25):6067-6080.e13. doi: 10.1016/j.cell.2021.11.009. Epub 2021 Nov 30. PMID: 34852238.

9) Overall the author should describe the environmental perturbants (endosomal pH, temperature etc) driving membrane fusion.